# Cyclic-di-GMP controls Type III effector export and symptom development in *Pseudomonas syringae* infections via the export ATPase HrcN

Danny Ward[1], Richard H. Little[1], Catriona M. A. Thompson[1], Jacob G. Malone [1,2]*

**1** John Innes Centre, Norwich Research Park, Norwich, United Kingdom, **2** University of East Anglia, Norwich Research Park, Norwich, United Kingdom

* Jacob.malone@jic.ac.uk

## Abstract

*Pseudomonas syringae* is a destructive bacterial pathogen that infects a wide variety of plants. Following apoplastic entry, *P. syringae* uses its type 3 injectisome (T3I) to secrete host-specific effectors into the cytoplasm, enabling tissue wetting and immune suppression and leading to bacterial proliferation, chlorosis and necrosis. *P. syringae* strains encode dozens of highly specialised effectors, whose composition defines strain specificity and host range. Effective plant infection depends on the tight temporal and hierarchical control of effector delivery through the T3I. Effector secretion is driven by HrcN, an ATPase complex that interacts with the base of the T3I and is essential for plant infection. HrcN binds specifically to the bacterial signalling molecule cyclic-di-GMP, although the impact of binding on T3I function and *P. syringae* virulence is currently unknown. To address this, we examined the influence of mutating the predicted cyclic-di-GMP-*hrcN* binding site on plant infection and effector secretion. Despite maintaining effective bacterial proliferation in *Arabidopsis thaliana* leaves, two *hrcN* mutants showed severely compromised disease symptoms, a phenotype linked to reduced translocation of a specific subset of T3I-effectors, with HopAA1–2 particularly important for symptom development. We propose that cyclic-di-GMP binding may represent a novel regulatory mechanism for effector secretion during bacterial infections.

## Author summary

*Pseudomonas syringae* is a bacterial plant pathogen that uses a molecular needle; the type 3 injectisome (T3I), to secrete specialised effector proteins into plant cells. Effector secretion through the T3I is vital for successful infection, enabling bacterial proliferation, immune system suppression and plant tissue degradation. HrcN is an ATPase protein that sits at the base of the T3I, and drives effector delivery through it. We previously showed that HrcN binds specifically to

**Data availability statement:** All data used in this submission are contained in the main manuscript and supplementary information files.

**Funding:** JGM, CMAT and RHL were funded by John Innes Centre which was funded by Biotechnology and Biological Sciences Research Council (BBSRC, https://www.ukri.org/councils/bbsrc/) Institute Strategic Programme grants BBS/E/J/000PR9797 and BB/X010996/1. DW was funded by a BBSRC Doctoral Training Partnership (BB/J014524/1) PhD studentship. The funders had no role in study design, data collection and analysis, decision to publish, or preparation of the manuscript.

**Competing interests:** The authors have declared that no competing interests exist.

the bacterial signalling molecule cyclic-di-GMP, although the purpose of this is currently unknown.

In this study, we use a combination of genetics, plant infection assays and biochemistry to investigate the role of cyclic-di-GMP in HrcN function. We show that mutations in the HrcN cyclic-di-GMP binding site can effectively decouple bacterial proliferation from disease symptoms during infection, producing rare asymptomatic plant infections. This phenomenon is due to reduced secretion of a subset of T3I-effectors in bacteria with mutated HrcN proteins. Of these compromised effectors, one in particular; HopAA1–2, is shown to be highly important for symptom development. Cyclic-di-GMP binding to HrcN may therefore represent a newly discovered mechanism for controlling effector secretion during *P. syringae* plant infections.

## Introduction

*Pseudomonas syringae* is a diverse species complex of Gram-negative foliar plant pathogens, comprising over 100 distinct pathovars [1] with the ability to infect a wide variety of plants, including almost all economically important crops [2]. *P. syringae* is a widely used model organism for the molecular analysis of plant–microbe interactions [2,3], with the tomato pathogen *P. syringae* pv. *tomato* DC3000 (*Pto*) particularly well-studied due to its ability to infect the model plants *Arabidopsis thaliana* and *Nicotiana benthamiana* [4,5]. Upon entry into the apoplast through surface wounds and leaf stomata, *P. syringae* infections proceed via tissue wetting accompanied by rapid bacterial proliferation, then chlorosis and eventual tissue necrosis [6].

A key organelle in the *P. syringae* infection process is the type 3 injectisome (T3I), with a functional T3I necessary for plant infection and apoplastic proliferation [7]. This large, multiprotein complex spans the inner and outer membranes, connecting the bacterial cytoplasm to that of its host and enabling the translocation of host-specific protein effectors into the host cytoplasm [8]. Much of our mechanistic understanding of T3I assembly and function is derived from studies of human pathogens such as *Escherichia coli, Salmonella* and *Yersinia*, and the secretory system of the bacterial flagellum [9–15]. However, the core T3I components share a high level of structural similarity and a close evolutionary lineage with each other [13], supporting working hypotheses for the functions of many *P. syringae* T3I components. T3I assembly is tightly regulated, with extracellular subunits exported through the central pore of the lengthening organelle [16].

Recruitment and secretion of type 3 effectors (T3Es) proceeds via the recognition of highly variable secretion signals at the effector N-terminus [17–19] in a chaperone-dependent or independent manner [20]. Effector proteins are then recruited to a sorting platform at the base of the T3I [19,20]. This sorting platform consists of a C-ring (HrcQ and HrpD), export apparatus (HrcRST, HrcU and HrcV)

and an ATPase complex (HrpE and HrcN) [10,12]. The HrcN ATPase is a central component of T3I function, whose deletion almost entirely abolishes bacterial cytotoxicity in the plant pathogen *Xanthomonas campestris* [21].

HrcN forms a dodecamer at the base of the sorting platform [22] and controls substrate entry into the export gate by ensuring that ATP hydrolysis occurs in concert with secretion. It also ensures efficient substrate unfolding in the export channel [11,23,24]. In *E. coli* and *Salmonella* species, T3I ATPase monomers comprise a central ATPase domain flanked by an N-terminal domain containing a 5 stranded β-sheet and a predominantly α-helical C-terminal domain [9,14,25]. The cryo-EM structure of the closely related *E. coli* ATPase EscN reveals an asymmetric multimer, suggesting a rotational catalysis mechanism similar to that of F1 ATPases [14].

T3Es encoded by *P. syringae* pathovars fulfil diverse roles in plant infection, facilitating bacterial colonisation while repressing the host immune response [1,26]. T3Es can trigger apoplastic wetting [27], manipulate host NAD(+) metabolism [28], suppress plant immune responses [29–33] or interfere with other traits including organelle function, cell development, nutrient distribution and metabolism [34]. *P. syringae* strains typically contain 15–35 well-expressed T3Es (~36 in *Pto* [35]), drawing from at least 58 distinct effector families [36,37]. The effector repertoire of an individual strain is highly specialised towards colonisation of its host plant, with these differences one of the main factors defining strains and pathovars [38].

T3E delivery in *Salmonella* is under tight temporal and hierarchical control [15], and there is some evidence for similar mechanisms in *P. syringae*. For example, effectors enabling apoplastic wetting show a response in the initial stages of infection [27], while early- and late-stage immune responses are targeted by different T3Es. AvrPtoB targets the PAMP immune receptor FLS2 for degradation [39], while HopM1 disrupts late-stage immunity by inducing degradation of the vesicle trafficking-related protein AtMIN7 [40]. Leaf chlorosis and necrosis are apparently enabled by targeted translocation of a few key T3I effector proteins, rather than with a larger fraction of the effector protein repertoire [41–43]. Coordinated effector delivery to the T3I is enabled by specific *hrp*-associated (*hpa*) chaperones [44]. Furthermore, T3I deployment and activity are also subject to extensive regulatory control [16,45].

Our previous research showed that the ubiquitous dinucleotide signalling molecule cyclic-di-GMP (cdG) binds specifically to the ATPase proteins of several bacterial secretory systems, including the flagellar protein FliI and HrcN from *Pto* [46]. CdG is associated with virulence and infectious persistence in many human and plant pathogens [47] and exerts its effects by binding to diverse protein and RNA targets, which in turn control an array of bacterial traits [48]. In *P. syringae*, cdG tightly controls the transition between motile, unicellular and communal, biofilm-forming lifestyles [49], with T3I-mediated virulence among the regulated traits [45]. CdG binding to HrcN-like ATPases appears to be widespread, with examples seen with type II, III, IV and VI secretory systems from various species [46,50–52]. At this stage, however, the impact of cdG-HrcN binding on T3I activity and *Pto* virulence is unknown.

To examine this relationship in more detail, we produced mutants in several *Pto hrcN* residues predicted to play a role in cdG-binding, and tested their impact on plant infection, protein function and T3I effector secretion. Two *hrcN* mutants, *G176A* and *G311A*, showed severely compromised visual disease symptoms, despite maintaining effective bacterial colonisation of *A. thaliana* Col0 leaves. This asymptomatic phenotype could be reversed by disrupting *Pto* intracellular cdG levels, and depends on a functional plant immune response. Using *cyaA*-fusion effector translocation assays [53], we observed significantly reduced translocation of a subset of T3I-effectors in the *G176A* mutant compared to WT *Pto*, with the asymptomatic phenotype linked to impaired HopAA1–2 secretion in particular.

## Results

### Specific HrcN mutations ameliorate disease symptoms upon *P. syringae* leaf infection

In our previous work we showed that cdG binds specifically to the HrcN ATPase, with dinucleotide binding sites predicted to lie at the interface between two monomers within the larger dodecameric complex at the base of the *P. syringae* T3I

[46]. At this stage, however, the biological function of this interaction is unknown. To address this, we first highlighted several residues predicted in our original study to contribute to the HrcN cdG binding site on an AlphaFold3 model of *P. syringae pv. tomato* DC3000 (*Pto*) HrcN (Fig 1A). The predicted positions of these residues were broadly in agreement with our earlier work (Fig 1B, upper panel) [46], with each predicted either to contribute directly (E208), or to coordinate the position of residues contributing to the dinucleotide binding pocket (F174, G176, G311, L338). The predicted cdG binding site is comprised of several arginine, aspartate and glutamate residues positioned at the interface between the two HrcN subunits (Fig 1B, lower panel), consistent with cdG binding motifs in other proteins [54].

Next, we constructed a series of mutants, introducing conservative substitutions of each selected residue into *Pto hrcN*, with the intention to either disrupt cdG binding, or to alter its downstream effects on protein function. We then tested the capacity or each mutant to infect Col0 *Arabidopsis* plants, relative to WT *Pto* and Δ*hrcC* & Δ*hrcN* negative control strains. Bacterial load was measured for each strain at 2 and 3 days post-leaf infiltration. The Δ*hrcC*/Δ*hrcN* negative controls showed little increase in bacterial load from the initial inoculum, consistent with the loss of T3I function (S1A Fig). Meanwhile, a 3–4 log increase in load was observed for WT *Pto* and all five tested mutants, indicating that the *hrcN* point mutants do not affect bacterial proliferation *in planta*. Immunoblotting of infected leaf tissue 3 days post-infiltration confirmed the presence of HrcN in all samples except for Δ*hrcC*, Δ*hrcN* and the uninfected plant tissue controls (Figs 1D, S1B). Assembly of the HrcC baseplate is thought to be necessary for HrcN production and structural stability [13], explaining the absence of HrcN in the Δ*hrcC* control sample.

Six days post-bacterial infiltration with WT *Pto*, infected leaves displayed systemic chlorosis and tissue necrosis, in contrast to the negative controls where no symptoms of infection were observed. Strikingly, however, while the disease symptoms for *hrcN*^E208D^, *hrcN*^L338V^ and *hrcN*^F174Y^ infections did not differ noticeably from WT *Pto*, minimal disease symptoms were observed for the *hrcN*^G176A^ and *hrcN*^G311A^ mutants, with little visible chlorosis or necrosis in either case (Fig 1C). Observed differences in leaf discolouration were quantified using an average pixel intensity calculation applied to each leaf image using ImageJ (version 1.52a), with highly significant differences in the proportion of discoloured leaf tissue observed between WT *Pto, hrcN*^G176A^ and *hrcN*^G311A^ (S1C Fig).

We then complemented the *hrcN*/*hrcC Pto* mutations by integrating a WT copy of *hrcN* (or *hrcC*) under the control of its native promoter into the neutral chromosomal *att*::Tn*7* site of each strain. Full recovery of tissue chlorosis and necrosis was observed upon leaf infection with the complemented *hrcN*^G176A^ and *hrcN*^G311A^ strains, suggesting that the asymptomatic infection phenotype resulted from the *hrcN* mutation in each case. Δ*hrcC* and Δ*hrcN* also displayed WT infection phenotypes when complemented, with no differences in infection phenotypes observed for the other complemented mutants (S1D-S1E Fig).

## The G176A mutation does not compromise HrcN ATPase activity or cdG binding

To attempt to probe the relationship between cdG-binding and HrcN function in more detail, we next expressed and purified N-terminal truncated, His6-tagged HrcN alongside G176A, G311A and a P142Q variant whose non-conservative residue substitution is predicted to disrupt protein folding. Removing the N-terminal 18 residues of *S. enterica* FliI improves *in vitro* protein stability [55], but abolishes higher-order multimerization [56]. *P. fluorescens* FliI Δ1–18 retains both ATPase activity and cdG binding ability [46], influencing our decision to purify truncated HrcN variants. Despite this, expression levels of all HrcN variants were low, likely due to heterologous host incompatibility. The G311A variant also expressed poorly and appeared to be slightly insoluble.

WT HrcN, HrcN^G311A^ and HrcN^G176A^ retained ATPase activity, unlike HrcN^P142Q^ and in line with our previous findings [46] (Fig 2A). Unfortunately, given the difficulty of working with HrcN^G311A^, we were forced to restrict our subsequent analysis to the G176A variant only. Next, we confirmed cdG binding to WT HrcN using DRaCALA assays [57] with fluorescent cdG (Fig 2B). Curiously, the HrcN^G176A^ variant also bound cdG (Fig 2B-2C), without apparent loss of binding affinity, in contrast to earlier results seen for *P. fluorescens* FliI [46]. CdG binding to both HrcN variants was specific; fluorescent cdG binding

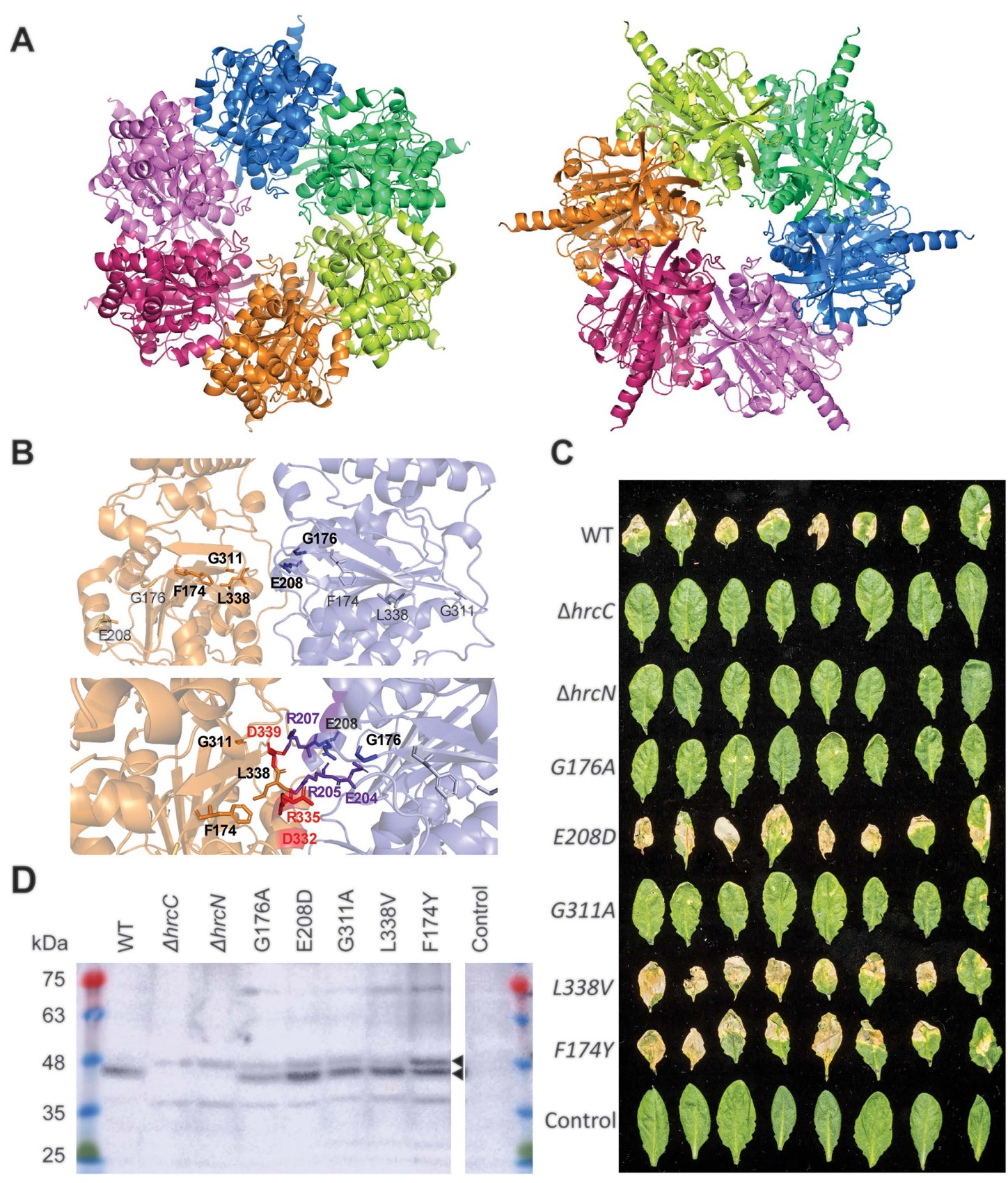

**Fig 1. (A) AlphaFold3 structural predictions for hexameric HrcN complexes.** Left: C-terminus facing; Right: N-terminus facing. **(B)** Upper panel: Residues mutated in this study, highlighted on the HrcN AlphaFold3 structure. Lower panel: Location of mutated residues (black font) surrounding predicted cdG binding residues in red (orange subunit) and purple (blue subunit). Note that each binding site is predicted to incorporate residues from two adjacent subunits. **(C)** *A. thaliana* Col-0 disease phenotypes at 6 days post-infiltration infection with *Pto* DC3000 strains containing deletions in *hrcC*/ *hrcN*, alongside mutations leading to residue substitutions surrounding the predicted *hrcN* cdG binding site and an uninfected control. Leaves taken at random from four independent plants are shown in each case. **(D)** Anti-HrcN (bottom) Western blot of *A. thaliana* Col-0 leaf tissue infected with *Pto* DC3000 containing *hrcC*/*hrcN* mutant alleles. Infected leaf tissue was collected three days post-infection. Colony forming units (CFU) recovered from infected leaves are quantified in S1A Fig. The location of HrcN (present in all samples apart from ΔhrcC and ΔhrcN) are indicated. FliI appears as a contaminating band with a slightly higher mass than HrcN. Both protein identities were confirmed by mass spectrometry.

could be disrupted by the addition of unlabelled cdG, but not by a range of other nucleotides (Fig 2C). Furthermore, neither HrcN variant could bind the fluorescent dinucleotide cyclic-di-AMP (cdA, Fig 2D).

## The *hrcNG176A* and *hrcNG311A* infection phenotypes are linked to bacterial cdG levels and plant immunity

To investigate the *hrcN*$^{G176A}$ and *hrcN*$^{G311A}$ asymptomatic infection phenotypes in more detail, we conducted additional rounds of leaf infection with strains over-expressing the well-characterised cdG synthase and phosphodiesterase genes *wspR*$^{19}$ and *bifA* [49,58], producing strains with elevated and suppressed cdG levels, respectively. CdG overproduction by the autoactive diguanylate cyclase variant WspR$^{19}$ fully recovered tissue chlorosis and necrosis in both *hrcN*$^{G176A}$ and *hrcN*$^{G311A}$ infections (Figs 3A, S2A-S2B), suggesting that excess cdG is able to override the effects of the *hrcN* mutation in each case. Curiously, symptom recovery was also observed for low-cdG, *bifA*-expressing strains, albeit only partially in the case of *hrcN*$^{G176A}$ (Figs 3B, S2C-S2D). Disruption of cdG levels did not produce noticeable effects on infections with any of the other tested mutants or control strains.

We next examined the impact of *hrcN* mutation on infections of immune compromised plants. Infection of Col0 *bbc* plants; missing the *bak1–5, bkk-1*, and *cerk1* immune receptor genes [27], with WT *Pto* and all five *hrcN* mutants led to strong infection phenotypes, in contrast to the ΔhrcC, ΔhrcN and water controls (Figs 3C, S3A-S3B). Similar results were seen with Col0 *fec* plants (with *fls2, efr* and *cerk1* immunity genes deleted [59]). Here, strong infection phenotypes were observed for WT *Pto*, *hrcN*$^{G176A}$, *hrcN*$^{E208D}$ and *hrcN*$^{G311A}$ in contrast to the ΔhrcC negative control (S3C-S3E Fig). These data suggest that the asymptomatic phenotype of *hrcN*$^{G176A}$ & *hrcN*$^{G311A}$ mutants may stem from reduced secretion of effector proteins that target the *Arabidopsis* immune system.

## *Pto hrcNG176A* and *hrcNG311A* mutants display compromised T3I effector secretion

To probe the relationship between the *hrcN*$^{G176A/G311A}$ infection phenotypes and effector secretion, we constructed CyaA-fusion vectors for 23 randomly selected *P. syringae* effectors [53]. These effector fusions are unfolded until they translocate into plant tissue, where translocation can be detected using ELISA assays for cAMP synthesis [53]. The vectors were transformed into WT *Pto*, ΔhrcN, *hrcN*$^{G176A}$ and *hrcN*$^{G311A}$, plants were infected, and relative effector translocation measured in each case (Fig 4A, 4B). Strikingly, both *hrcN*$^{G176A}$ and *hrcN*$^{G311A}$ showed substantial differences in the extent of translocation relative to WT *Pto* for multiple effectors. No effector translocation was observed for any ΔhrcN strain, as expected.

Due to the low sample sizes used in these initial screening experiments, the results in Fig 4A and 4B showed a high degree of error. We therefore selected four effectors for more detailed analysis. HopAA1–2 and HopAM1 showed the greatest translocation reductions in *hrcN*$^{G176A}$ and *hrcN*$^{G311A}$ respectively and were both strongly reduced in *hrcN*$^{G176A}$, while HopAF1 showed strongly reduced translocation in *hrcN*$^{G311A}$ and mildly reduced translocation in *hrcN*$^{G176A}$. HopH1 plays a role in *Pto* virulence in *Arabidopsis* [60], but showed little difference in translocation across any of the tested strains and was included as a negative control. Effector translocation assays with these four proteins were then conducted in WT *Pto,* ΔhrcN and *hrcN*$^{G176A}$. In line with our previous findings, absolute levels of effector secretion from *hrcN*$^{G176A}$ were

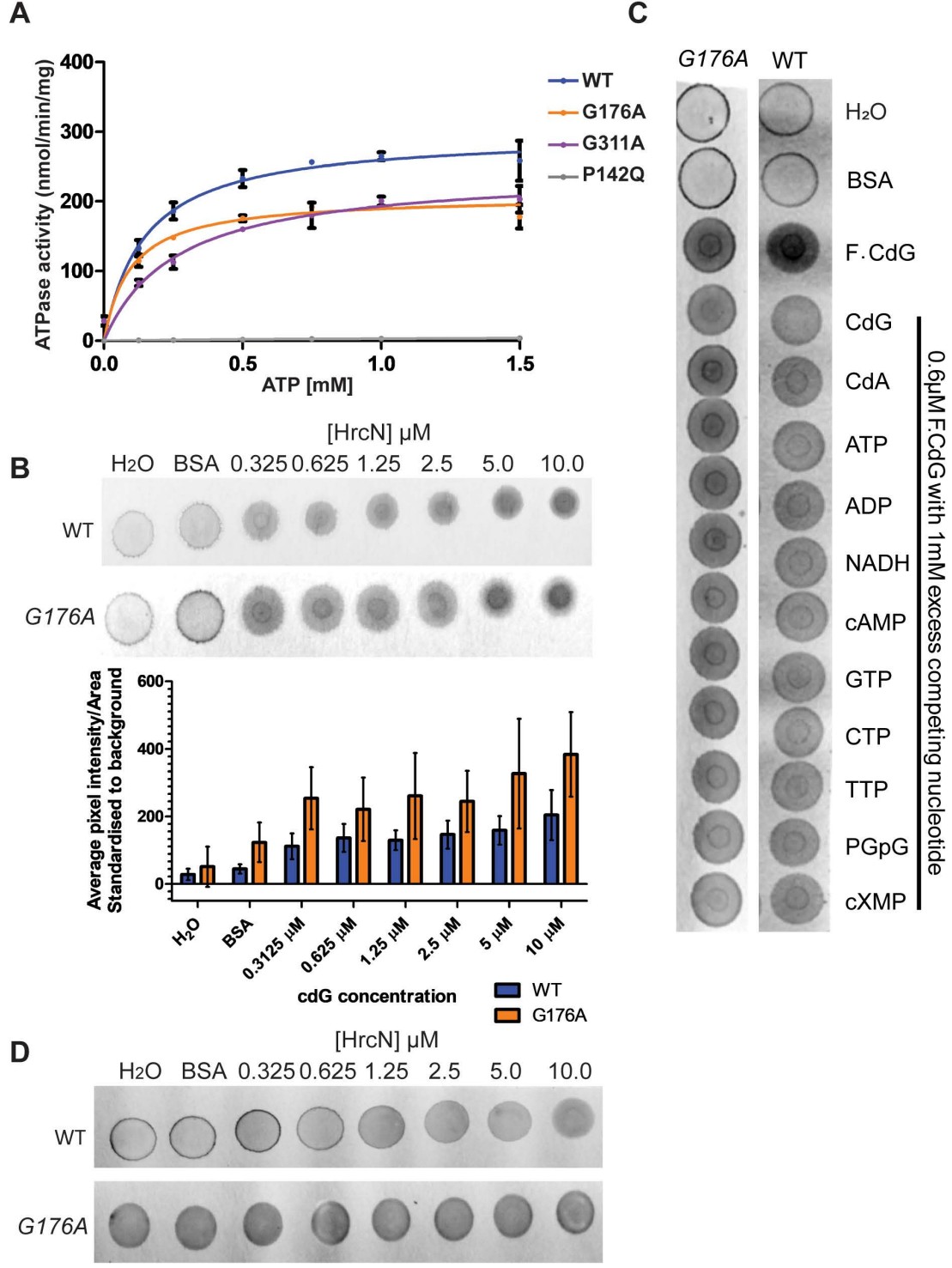

**Fig 2. (A) ATPase activity demonstrated for purified Δ1-18 HrcN variants using a PK/LDH ATPase assay**. This reaction has been plotted as a Michaelis-Menten saturation curve where [HrcN] is 1 μM. Error bars show standard deviation. **(B)** Upper panel: Fluorescent-CdG DRaCALA assays with increasing concentrations of purified Δ1-18 WT HrcN (top) and Δ1-18 HrcN$^{G176A}$ (bottom). Fluorescent-CdG is included at a fixed 0.6 μM final concentration. Representative assays are shown in each case. Lower panel: ImageJ quantification of HrcN DRaCALA datasets (n = 3). No significant differences were observed in binding affinity between WT and G176A HrcN variants. Error bars show standard deviation. **(C)** Competition DRaCALA

assays with excess unlabelled nucleotides. Δ1-18 WT HrcN/ HrcN$^{G176A}$ and fluorescent-CdG are present at fixed final concentrations of 1.25 μM and 0.6 μM, respectively. Non-labelled competitor nucleotides, as indicated, were added to a final concentration of 1mM. **(D)** Fluorescent-CdA DRaCALA assays with increasing concentrations of purified Δ1-18 WT HrcN and Δ1-18 HrcN$^{G176A}$. Fluorescent-CdA is included at a fixed 0.6 μM final concentration. B-D: Control samples contain either H$_2$O or 10 μM BSA in place of HrcN, as indicated.

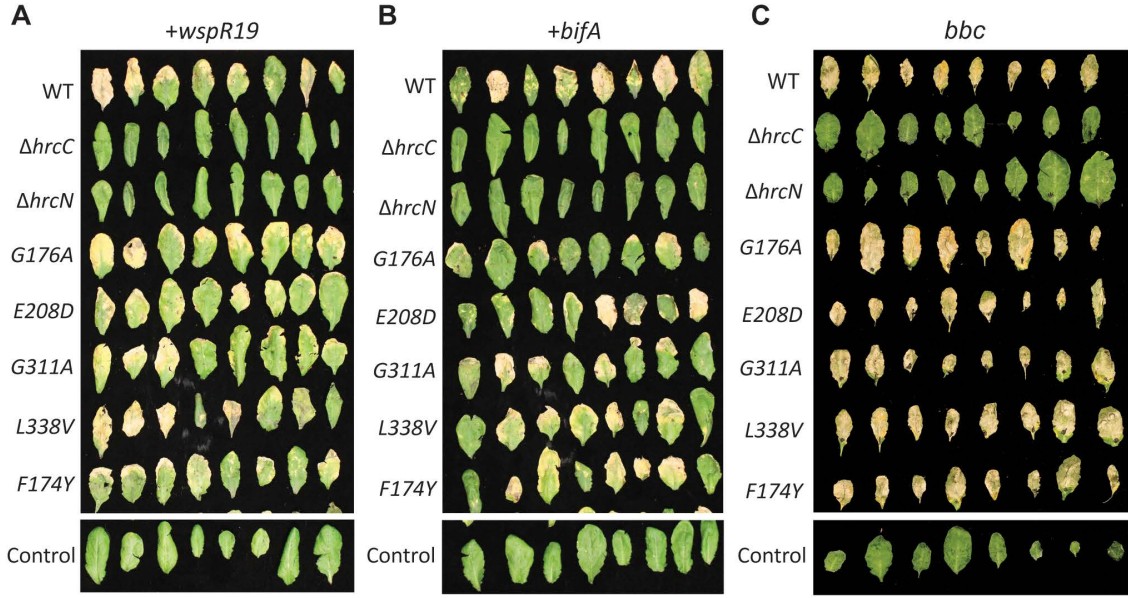

**Fig 3. (A-B) Visual disease phenotypes of *Pto* DC3000 infiltrated *A. thaliana* Col-0 leaves, 6 days post-infection.** *Pto* strains harbour mutant *hrcC/hrcN* alleles as indicated, alongside **A:** pBBR4-*wspR19* (inducing high cdG levels) and **B:** pBBR4-*bifA* (inducing low cdG levels). **(C)** Disease phenotypes of *Pto* DC3000 infiltrated *A. thaliana* bak1-5,bkk-1,cerk1 (*bbc*) leaves, 6 days post-infection. *Pto* strains harbour mutant *hrcC/hrcN* alleles as indicated. Imaged leaves were taken at random from three independent plants in each case.

significantly lower than WT *Pto* for HopAA1–2, HopAM1 and HopAF1, while no significant difference was observed for HopH1 (Fig 4C). When measured over the course of 24 hours post-infection, the onset of HopAA1–2 translocation was delayed by approximately 3 hours in the *hrcN*$^{G176A}$ mutant compared to WT *Pto*, and occurred to a significantly reduced extent. No differences were seen for HopH1 translocation over the same time period (Fig 4D).

### *hrcNG176A*-dysregulated effectors contribute to the onset of infection phenotypes

To determine the impact of reduced effector expression on *Arabidopsis* infection, *hopAA1–2, hopAM1, hopAF1,* and *hopH1* were expressed *in trans* in WT *Pto* and *hrcN*$^{G176A}$ and their impact on infection phenotypes was assessed (Fig 5). Overexpression of *hopAA1–2, hopAM1* and *hopAF1* had little or no impact on infection phenotypes (Fig 5A) or bacterial load (S4A-S4B Fig) in WT *Pto*, but led to a partial recovery of chlorosis and necrosis in *hrcN*$^{G176A}$. Conversely, overexpression of *hopH1* had little effect on infection phenotypes with either tested strain.

Finally, we examined the impact of deleting the compromised effectors on Col0 infections by WT *Pto*. While our efforts to generate Δ*hopAF1* were unsuccessful, we were able to produce *Pto* Δ*hopAA1–2*, Δ*hopAM1* and a Δ*hopAA1–2/hopAM1* double mutant. Strikingly, while *hopAM1* deletion produced no obvious effects on *Pto* infection phenotypes, both *hopAA1–2* deletion mutants phenocopied the *hrcN*$^{G176A}$ strain, with a loss of infection phenotypes (Fig 5B) but no accompanying reduction in bacterial load (S4C-S4D Fig). Together, these data strongly suggest a central role for HopAA1–2 in enabling symptom onset during *Pto Arabidopsis* infections.

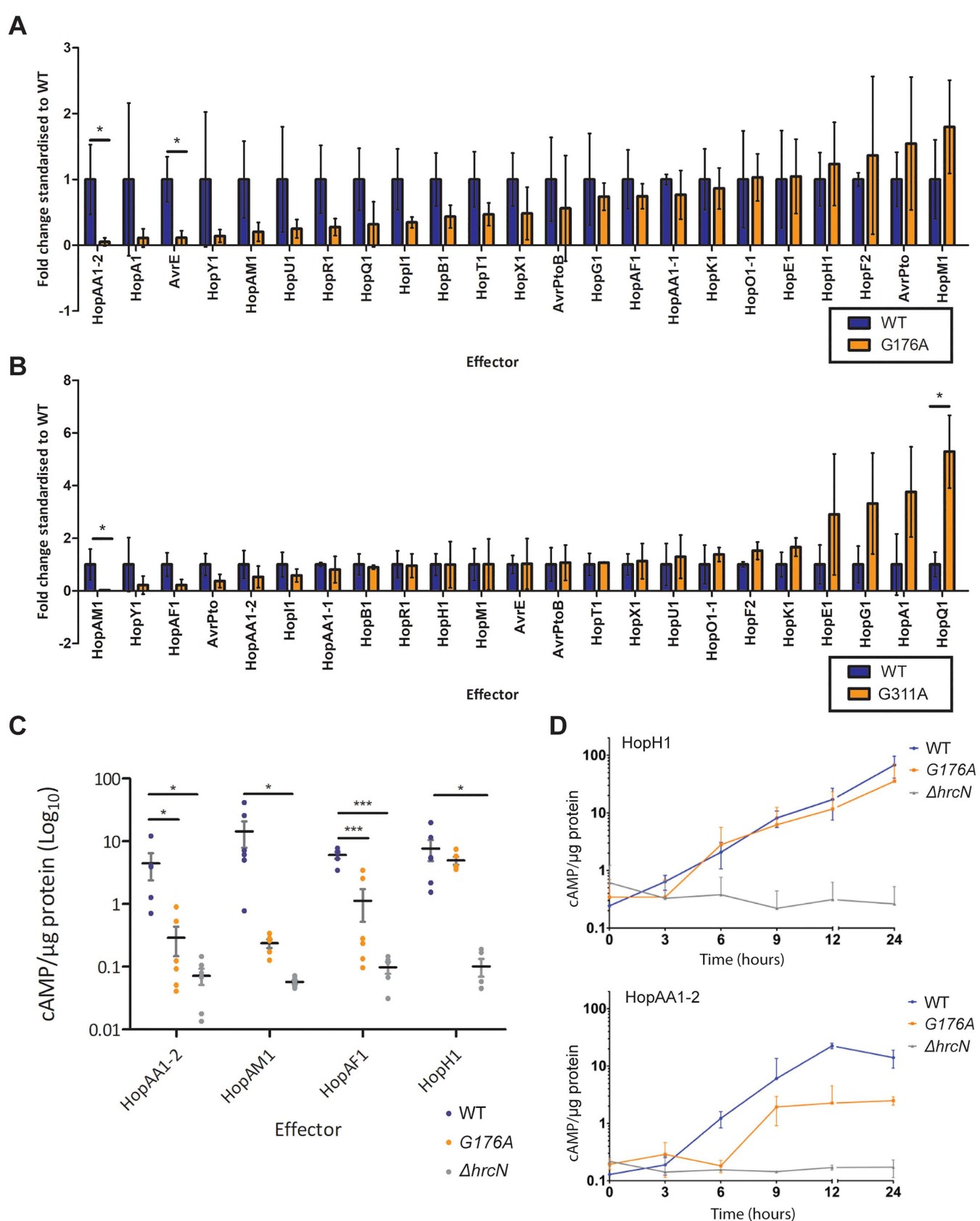

**Fig 4. (A)** *Pto hrcN*<sup>G176A</sup> effector-CyaA translocation rates shown relative to WT *hrcN*. **(B)** *Pto hrcN*<sup>G311A</sup> effector-CyaA translocation rates shown relative to WT *hrcN*. In each case, values above 1 indicate an increase in translocation for a given effector in the *hrcN*<sup>G176A</sup>/hrcN<sup>G311A</sup> mutant compared to WT, values below 1 show a decrease. Values compared were cAMP/µg protein (normalised against non-specific binding controls) in *Pto* infected *A. thaliana* Col-0 leaves 6 hours post-infection, where cAMP/µg protein directly correlates with effector translocation. Asterisks (*) denote statistical significance compared to the mean WT value, where '*' = p ≤ 0.05 (unpaired two-tailed t-tests, n = 3). **(C)** Effector-CyaA fusion reporter assays for key *Pto* effectors showing cAMP/µg protein (normalised against non-specific binding controls) in *Pto* DC3000 infected *A. thaliana* Col-0 leaves 6 hours post-infection, where cAMP/µg protein directly correlates with effector translocation. In each case, asterisks (*) denote statistical significance (one-way ANOVA with Tukey's multiple comparisons displayed, n = 6) compared to the mean WT value, where '*' = p ≤ 0.05 and '***' = p ≤ 0.01. **(D)** Effector-CyaA fusion reporter assays showing cAMP/µg protein in *Pto* DC3000 infected *A. thaliana* Col-0 leaves over 24 hours post-infection. Top graph: HopH1 effector translocation, bottom graph: HopAA1-2. *Pto* strains harbour mutant *hrcN* alleles as indicated.

## Discussion

The coordinated deployment of T3Es is likely to play an important role in effective T3I-mediated infections of both plants and animals [6,15]. In addition to regulating T3I deployment and activity [16,45], and controlling the delivery of specific T3Es to the injectisome [44], our findings suggest that secretion of specific T3Es during *P. syringae* plant infections is also controlled by allosteric regulation of T3I function, linked to the second messenger cdG. Conservative point mutations in residues linked to *hrcN* cdG binding [46] led to markedly altered virulence phenotypes upon *A. thaliana* infection, with *Pto hrcN*<sup>G176A</sup> and *hrcN*<sup>G311A</sup> mutants showing few visible signs of disease, despite bacterial apoplastic proliferation remaining unaffected.

The asymptomatic infection phenotypes seen for the *Pto hrcN*<sup>G176A/G311A</sup> mutants are unlikely to be the result of compromised T3I assembly or a nonspecific reduction in effector secretion. Bacterial proliferation in the apoplast requires a functional T3I [7], and HrcN was readily detected *in planta* in both *hrcN* mutants. Furthermore, ATPase activity remains intact in both HrcN<sup>G176A</sup> and HrcN<sup>G311A</sup>, and secretion of many T3Es is unaffected relative to wild-type *Pto*. Together, these results strongly argue for the presence of fully assembled, if compromised, T3Is in the *hrcN*<sup>G176A/G311A</sup> mutants.

While the majority of effectors are secreted normally in both mutants, *Pto hrcN*<sup>G176A</sup> and *hrcN*<sup>G311A</sup> each show disrupted export of a specific subset of T3Es, including several responsible for the onset of disease symptoms such as tissue chlorosis and necrosis. For the HopAA1–2 effector, compromised export persists over time, with levels associated with *hrcN*<sup>G176A</sup> markedly lower than for WT over a 24 hour time course (Fig 4D). This result suggests that export of at least some T3Es is permanently reduced, rather than delayed in the *hrcN*<sup>G176A</sup> and *hrcN*<sup>G311A</sup> mutants. Immunocompromised plants showed wild-type infection symptoms upon infiltration with both *hrcN* mutants, suggesting that the disrupted effectors function by suppressing elements of the plant immune response, which in-turn counter the destruction of plant tissues during infection.

At this stage, the mechanistic relationship between the HrcN mutants, cdG binding and T3I regulatory control remains unclear. It is possible that the slight loss of ATPase activity seen for the HrcN variants in vitro contributes to altered T3I selectivity. However, Minamino and coworkers showed that only residual ATPase activity is needed for flagellar injectisome functionality [23], and proton motive force can bypass the requirement for FliI ATPase activity entirely [24], arguing against this hypothesis. We were unable to confirm cdG binding to HrcN<sup>G311A</sup>, however, binding to HrcN<sup>G176A</sup> was not significantly different to that seen for WT HrcN. This is not entirely surprising, as this residue does not directly contribute to the predicted cdG binding interface (Fig 1C). Nonetheless, it suggests that the G176A phenotype may not result from reduced cdG binding affinity directly but instead represents a partial decoupling of cdG binding from its downstream impacts on T3I function. How this takes place is currently unknown, but may involve the disruption of HrcN multimerization, downstream allosteric signaling to T3Es or other secretion targets, or altered interactions with other T3I components.

Wild-type infection symptoms were recovered in both HrcN mutants upon diguanylate cyclase expression. This suggests that high levels of cdG may compensate for reduced HrcN binding affinity or inefficient signal transduction,

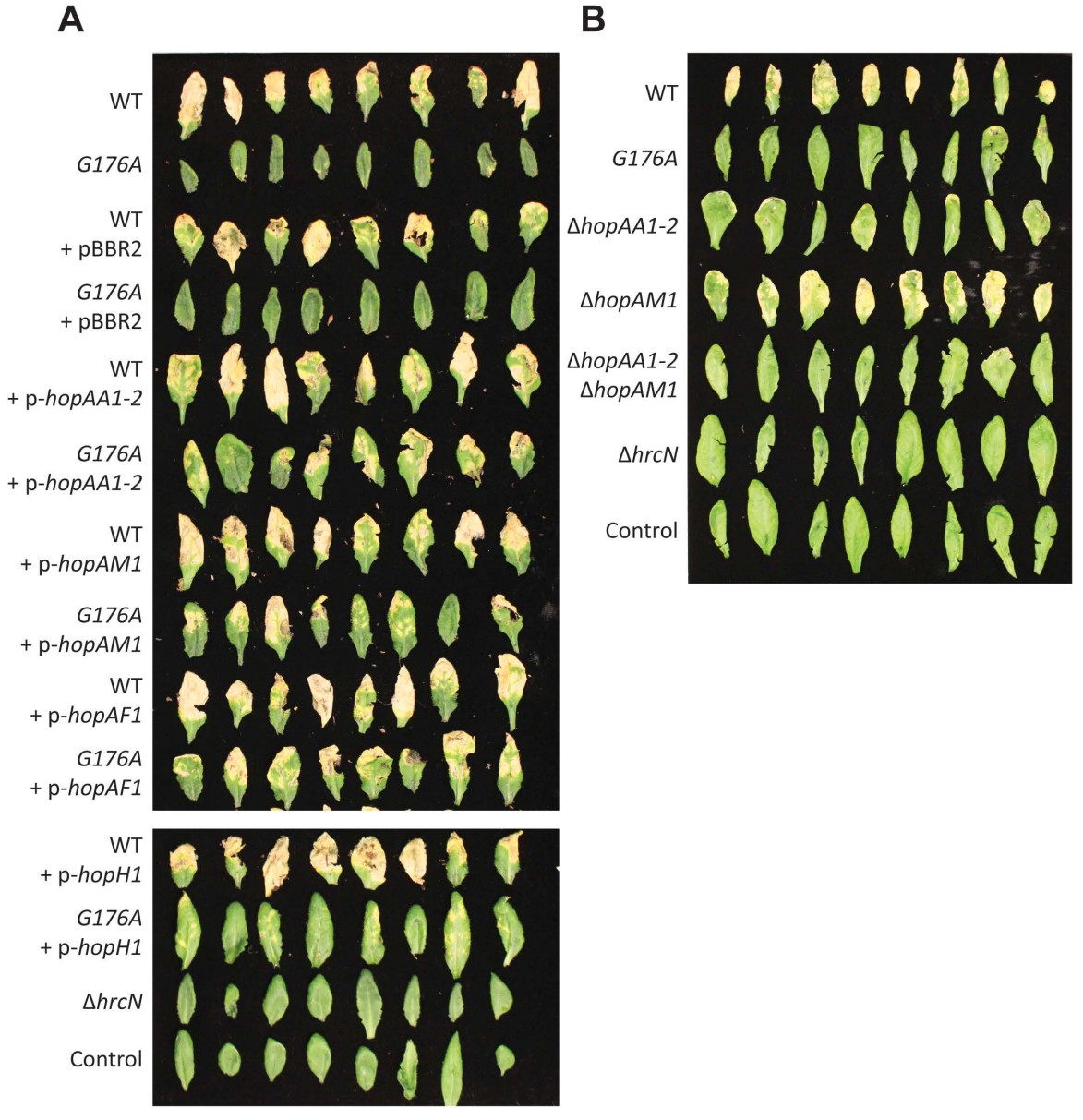

**Fig 5. (A-B) Visual disease phenotypes of *Pto* DC3000 infiltrated *A. thaliana* Col-0 leaves, 6 days post-infection. A:** *Pto* strains harbour the *hrcN*^G176A allele/Δ*hrcN* and effector expression plasmids as indicated. pBBR2: empty vector. **B:** *Pto* strains harbour the *hrcN*^G176A allele or effector/*hrcN* deletions as indicated. In each case, control indicates uninfected leaves, and imaged leaves were taken at random from three independent plants.

overcoming the reduced responsiveness of the G176A/G311A mutants. The observed symptom recovery upon phosphodiesterase expression is less clear-cut, but suggests that reduced cdG levels may exert indirect, positive impacts on *Pto* infectivity. An intriguing possibility is raised by the related human pathogen *P. aeruginosa*, where T3I production is controlled by cdG, with low levels required for production [52,61]. If *Pto* behaves similarly, then reduced T3I activity in the HrcN mutants might be countered by a general increase in T3I/T3E production upon phosphodiesterase expression. Further research is needed to fully dissect the relationship between cdG and HrcN complex formation in *P. syringae*.

The decoupling we observe between leaf disease symptoms and bacterial proliferation in *P. syringae* infections appears to be a rare phenomenon, with changes to disease symptom severity typically coupled to a corresponding increase or decrease in apoplastic colonisation [62,63]. Symptomless *P. syringae* infections have previously been reported where the induction of HR in non-host plants was missing [64]. Decoupling of disease severity and colonisation has also been reported for mutants in phytotoxin pathways. *Pto* strains lacking syringolin A undergo asymptomatic infections of wheat plants, whereas bacterial colonisation is relatively unaffected [65]. A similar reduction in necrotic lesion formation has been also reported for strains missing the phytotoxin coronatine, albeit in this case accompanied by a slight drop in bacterial load [62].

HopAM1, HopAF1 and particularly HopAA1–2 appear to play key roles in mediating *Pto* symptom development, and their export is post-transcriptionally regulated by HrcN. Using ELISA analysis of T3E-CyaA reporter fusion strains, we showed that delivery of specific effector proteins into leaves, including HopAA1–2, HopAM1, and HopAF1 was significantly compromised in the $hrcN^{G176A/G311A}$ mutants relative to wild-type. While over-expression of all three effectors partially recovered disease phenotypes in infections with the asymptomatic *Pto* $hrcN^{G176}$ mutant, only *hopAA1–2* appeared to be necessary for full establishment of disease phenotypes, with *hopAM1* deletion mutants showing similar visual disease symptoms to wild-type *Pto*.

HopAA1–2, HopAM1 and HopAF1 have all previously been linked to immune system dysregulation, which aligns with our observation of symptom recovery for $hrcN^{G176A}$ infections in immune compromised plants. HopAA1–2 interacts with EDS1 (enhanced disease susceptibility 1) and PBS3, and is hypothesised to play a role in salicylic acid-mediated defence subversion [66]. EDS1 is involved in transcriptional mobilisation associated with resistance pathways [67], while PBS3 protects EDS1 from proteasome-mediated degradation [68]. Reduced levels of both genes were seen upon co-expression with HopAA1–2 [66]. Deletion of a gene cluster containing *hopAA1–2* (alongside *hopV1*, *hopAO1, and hopG1*) showed strongly reduced *Pto* virulence in *N. benthamiana* and *A. thaliana* plants [60].

HopAM1 shares high Toll/interleukin-1 receptor (TIR) domain homology to NB-LRR receptors, which can hydrolyse nicotinamide adenine dinucleotide ($NAD^+$) [69]. It has been shown to induce meristem chlorosis and local cell death in *Arabidopsis* [70]. EDS1, alongside HSP90.2 and SGT1b; part of a steady-state NLR chaperone complex were shown by genome-wide association mapping to affect HopAM1-induced cell death [70]. HopAM1 has also been linked to virulence enhancement in water-stressed plants by inducing hypersensitivity to abscisic acid [71].

Finally, HopAF1 can suppress plant immunity by blocking ethylene induction through targeting of methionine recycling [72]. HopAF1 showed an interaction with Arabidopsis methylthioadenosine nucleosidase (MTN1 and MTN2) in yeast 2-hybrid screens [72,73]. MTN enzymes are known to be involved in the Yang cycle, a process essential for high levels of cellular ethylene in *A. thaliana* [72]. HopAF1 has also been shown to suppress production of defensive reactive oxygen species in the phytopathogen *Pseudomonas savastanoi* [74].

Bacterial cdG signalling relays diverse environmental inputs into integrated phenotypic responses [48]. This control can take place both at the level of transcription, but also by rapidly influencing the activity of existing proteins in the cell, as seen here. We propose that cdG binding to HrcN enables the allosteric control of T3E export for a subset of effectors involved in disease symptom development. Our model suggests that the coordinated deployment of bacterial virulence factors depends not only on transcriptional hierarchy and bacterial proximity to apoplastic cells, but is controlled actively and rapidly in response to signals from the apoplastic environment.

## Materials and methods

### Bacterial strains and growth conditions

Unless otherwise stated *P. syringae* strains were grown at 28°C and *E. coli* at 37°C with shaking. Bacterial strains and plasmids used in this study are listed in S1 Table. King's B medium (KB) [75] and L medium [76] were used for bacterial

cell culture as specified. Antibiotics were used at final concentrations of rifampicin at 50 µg/ml, nystatin at 25 µg/mL, tetracycline at 12.5 µg/ml, kanamycin at 50 µg/mL and chloramphenicol at 25 µg/mL, unless otherwise stated.

## Molecular cloning

Cloning was performed following standard molecular biology techniques [76]. *Pto* gene deletions and *hrcN* mutants were generated by allelic exchange, using vectors constructed by amplifying the upstream and downstream flanking regions of each mutagenesis site with primers DC3000-HrcN-FWD and DC3000-HrcN-REV plus appropriate internal point mutant primers (e.g., *Mut*-FWD/REV-DC3000-HrcN, see S2 Table), combining them by ligation/strand overlap extension PCR as appropriate and cloning into the suicide vector pTS1 between restriction sites *Xho*I and *Bam*HI [77]. *Pto* cells were transformed by electroporation with the appropriate deletion/mutagenesis vectors following the method described in [78]. Single crossover integrations of each plasmid into the chromosome were selected on tetracycline plates and re-streaked before single colonies were grown overnight in KB media without selection. Following a tetracycline enrichment step after [79], double crossovers were counter-selected by plating serial dilutions onto L agar containing 10% (w/v) sucrose. Deletion/site-specific *hrcN* mutants were subsequently confirmed by PCR and sequencing using primers FWDExternalHrcN and REVExternalHrcN (S2 Table).

To complement *Pto hrcN*/*hrcC* mutations with WT gene copies, the *hrcN*/*hrcC* genes and ~200 bp of upstream DNA incorporating the promoter region were amplified using the HrcNComp FWD and REV primers (S2 Table) and cloned into pUC18-miniTn7-Gm between *Hind*III and *Bam*HI/*Pst*I. The resulting vectors were transformed into *Pto* mutant strains by electroporation alongside the helper plasmid pTNS2 [80]. Chromosomal integration at the *att*::Tn7 site for successful transformants was verified by PCR using primers pTN7R and pGlmS-Down [80]. HrcN purification vectors were constructed by amplifying the *hrcN* alleles synthesised above using primers OE_DC3000_FWD and OE_DC3000_REV plus relevant internal point mutant primers (S2 Table) and cloning the resulting fragments between the *Nde*I and *Xho*I restriction sites of a pETM11 expression vector. The effector-CyaA fusion constructs were produced by gateway cloning of effectors and plasmids, as described previously [53]. pCPP5371-CyaA backbone vectors were Gateway cloned with T3E (lacking stop-codons)-pENTR-SD/D-TOPO vectors via an LR reaction to create pCPP5371::T3E + C-terminal CyaA vectors (pDEST) [53]. Effector protein overexpression vectors were produced by amplifying effector genes using primers, e.g., OvExp HopAF1 FWD/REV (see S2 Table for complete list) before cloning the resulting fragments between the *Kpn*I *and Xho*I restriction sites of a pBBR1MCS-2 expression plasmid [81].

## Protein purification

Protein purification was conducted after [79]. Briefly, overnight cultures of *E. coli* BL21-(DE3) pLysS containing pETM11-*hrcN*-His$_6$ vectors were used to inoculate 2.0 litres LB media (1:50 dilution). These cultures were then grown at 37°C, 220 rpm to an OD$_{600}$ of 0.4, before protein expression was induced overnight at 18°C, 220 rpm with 0.5 mM IPTG. Cells were harvested at 5000 g, 4°C for 10 minutes and re-suspended in 30 mL of ice cold chromatography binding buffer (20 mM HEPES, 250 mM NaCl, 2 mM MgCl$_2$, 3% v/v glycerol pH 7.8) supplemented with 1 mg/ml lysozyme, 10 µl DNaseI (Promega) and 1 complete protease inhibitor tablet EDTA-free (Roche), before immediate lysis on ice using an MSE Soniprep 150 ultrasonic disintegrator (15s on, 45s off, 15 cycles). The insoluble fraction was removed by centrifugation (18,000 x*g*, 35 minutes 4°C). The soluble fraction was loaded onto a HisTrap excel (Cytiva) column equilibrated with binding buffer. Following protein immobilisation, a column wash with 10% elution buffer (20 mM HEPES, 250 mM NaCl, 2 mM MgCl$_2$, 3% v/v/ glycerol, 500 mM imidazole pH 7.8) was performed to remove non-specific contaminants. Proteins were eluted over a linear gradient of 10–100% elution buffer. Protein purity was confirmed by SDS-PAGE and concentrations were calculated using Bradford assays [82]. Pooled fractions were desalted using PD-10 (Cytiva) according to the manufacturer's instructions, with final samples being stored at 4°C in storage buffer (20 mM HEPES, 250 mM NaCl, 2 mM

MgCl$_2$, 3% *v/v* glycerol pH 7.8). Proteins remained stable at 4°C for up to 5 days and were taken forward immediately for *in vitro* analysis.

### Protein structural predictions

The hexameric structure of HrcN was modelled by AlphaFold3 [83], using the default settings and specifying 6 monomeric subunits (ipTM = 0.78, pTM = 0.8). The downstream analysis and image generation was performed in Pymol.

### Linked pyruvate kinase/ lactate dehydrogenase (PK/LDH) ATPase activity assays

ATPase activity was measured indirectly by monitoring NADH oxidation in a CLARIOstar plate reader (BMG Labtech) at 25 °C. The reaction buffer consisted of 100 mM Tris-Cl (pH 8.0), 2 mM MgCl$_2$. Each 100 µL reaction contained 1 µM (final concentration) of purified HrcN, 0.4 mM NADH, 0.8 mM phosphoenolpyruvic acid, 0.7 µL PK/LDH enzyme (Sigma) and was initiated by the addition of 10 µL ATP solutions, 2 replicates of each concentration: 0, 0.125, 0.25, 0.5, 0.75, 1.0, 1.5, 2.0 mM ATP. Enzyme kinetics were determined by measuring A$_{340}$ at 90 second intervals. Kinetic parameters were calculated by plotting the specific activity of the enzyme (nmol ATP hydrolyzed/min/mg protein) versus [ATP] and by fitting the non-linear enzyme kinetics model (Michaelis-Menten) in GraphPad Prism.

### DRaCALA nucleotide binding assays

CdG/CdA binding assays were performed as described by Roelofs et al. [57]. Protein samples were incubated for 2 minutes at room temperature with 2'-Fluo-AHC-c-diGMP (BioLog 009) or 2'-Fluo-AHC-c-diAMP (BioLog C 088), in a final concentration of 0.6 µM in reaction buffer (25 mM Tris, 250 mM NaCl, 10 mM MgCl$_2$, 5 mM β-mercaptoethanol pH 7.5). For the competition experiments, 1 mM of each competing nucleotide was mixed with 1.25 µM HrcN and incubated for 30 minutes prior to the addition of the fluorescent di-nucleotides. The tubes were incubated on ice for 10 minutes in the dark. 5 µL of each sample was then spotted on nitrocellulose membrane and left to briefly dry before visualisation using the UV fluorescence mode on a G:Box F3 Imager (Syngene). Average pixel intensity was measured in ImageJ, with the mean pixel intensity subtracted from the background intensity and divided by the area measured. Analysis was performed for three independent replicates.

### Plant growth conditions

Col0 *A. thaliana* plants and immuno-compromised fec (*fls2, efr,* and *cerk1* gene knock-out) and bbc (*bak1–5, bkk1–1,* and *cerk1* gene knock-out) lines [84] were used in this study. Plants were grown in controlled environment horticultural facilities under short day (10-hour) lighting conditions [120–180 µmol m$^{-2}$ s$^{-2}$ light intensity], 70 percent relative humidity, and 22°C growth temperature. All seedlings were grown prior to infection for 4–5 weeks + 1 week 4°C seed vernalisation in mixed *Arabidopsis* peat (Levington F2 600 ITS Peat, 100 ITS 4mm grit, 196g Exemptor Chloronicotinyl Insecticide). Plant experiments were conducted in containment level 2 controlled environment rooms, under plant health licence 51103/197332/5.

### Plant infection assays

Cell cultures were grown overnight in L medium, harvested by centrifugation, then resuspended in 10mM MgCl$_2$ and adjusted to a final density (OD$_{600}$) of 0.0002 (equivalent to 10$^7$ cells ml$^{-1}$). Cells were gently infiltrated into the leaf apoplast using a 1 ml syringe until leaves appeared wet. Post infiltration, bacterial suspensions were allowed to absorb/dry onto leaf surfaces for approximately 1 h before initial (0 DPI) samples were collected. For calculating bacterial load, two 7mm diameter leaf discs (area 0.384 cm$^2$) were collected for each sample at 0, 2 and 3-days post-infection, flash-frozen, and ground using a Qiagen Tissue Lyser II (30 hz, 45 seconds x 2). Lysate was serially diluted 10$^{-1}$ to 10$^{-5}$ and plated onto LB

with rifampicin and nystatin. Plates were incubated for 2 days at 28 °C prior to colony counting. Bacterial load was calculated and presented as colony forming units per unit leaf area (CFU/cm$^2$). A minimum of three plants were used for each condition and assays were run at least twice independently.

## Western blotting of leaf tissue

Two 7mm discs of *Pto*-infected leaf tissue per sampled plant (3 days post-infection) were incubated with 100 µl SDS-PAGE sample buffer (BioRad) at 100°C for 10 minutes, before the supernatant was collected following centrifugation (16,000 g, 10 minutes, 4°C) to remove the insoluble plant material. Samples were then separated by SDS-PAGE gels and transferred onto a polyvinylidene difluoride (PVDF) membrane (Millipore). Membranes were incubated overnight in blocking solution at 8°C (1x PBS pH 7.4, 0.01% Tween-20, 5% w/v milk powder). Target protein was subsequently detected with 1/10,000 anti-HrcN primary antisera (Minotech Biotechnology) and 1/3000 anti-rabbit secondary antibody (Sigma). Western blots were visualized using ECL chemiluminescent detection reagent on an ImageQuant LAS-500 scanner (GE Healthcare). To verify the identity of non-specific antibody targets, SDS gel slices containing protein samples of interest were washed, treated with DTT and iodoacetamide and digested with trypsin according to standard procedures. Peptides were extracted from the gels and analysed by LC-MS/MS on an Orbitrap Fusion Tribrid Mass Spectrometer (Thermo Fisher, Hemel Hempstead, UK).

## ImageJ analysis

The level of chlorosis severity of infected leaves was quantified by average pixel analysis using ImageJ [85]. Images were opened and the leaf surface area was selected using the multi-point tool for each leaf. The measure setting was used to calculate pixel intensity (brightness of a given pixel) over the selected area on a scale of 1–255 (1 showing least pixel intensity and no evidence of chlorosis, 255 showing highest pixel intensity with strong evidence of chlorosis). A minimum of 8 leaves per sample were used to calculate final averages per replicate.

## CyaA effector fusion reporter assay

Pto strains containing effector-CyaA fusion constructs (in pCPP5295 stable expression vectors, driven by a *hrp*-promoter) were grown overnight in KB medium and diluted to an OD$_{600}$ of 0.05 (5 x 10$^7$ cfu/mL) in 10 mM MgCl$_2$ before infiltration into 4-week-old Col0 leaves as described above. 6 hours post-infection, 4 mm diameter leaf discs from the infected leaves were collected using a leaf corer, flash frozen in liquid nitrogen and ground using a Qiagen Tissue Lyser II (30 hz, 45 seconds x 2). 300 µL of 0.1 M HCl was added to each sample, samples were pelleted by centrifugation (13,000 g, 10 min), vortexed and then pelleted by centrifugation for a further 10 minutes. The supernatant was collected and diluted 1:5 and 1:50 in 0.1 M HCl. A direct cAMP ELISA kit (Enzo) was used to quantify [cAMP] in each sample following the manufacturer's instructions. Data was analysed by comparing each well against the logarithmic curve generated from the cAMP standards. This value was then divided by the calculated sample protein concentration (via standard Bradford assay) to give pmol cAMP/µg protein.

## Supporting information

**S1 Table. Strains and plasmids used in this study.**
(DOCX)

**S2 Table. Primers used in this study.**
(DOCX)

**S1 Fig. A.** Colony forming units (CFU) recovered from *A. thaliana* Col-0 leaves infiltrated with *Pto* DC3000 *hrcC*/*hrcN* mutants at 0, 2 and 3 days post-infection (DPI) as indicated. **B.** CFU recovered from *A. thaliana* Col-0 leaves infiltrated

with *Pto* DC3000 *hrcC/hrcN* mutants after 1 and 3 DPI. 3DPI samples were used for anti-HrcN Western blotting in Fig 1D. **C.** Average pixel intensity analysis for *A. thaliana* Col-0 leaves infiltrated with *Pto* DC3000 *hrcC/hrcN* mutants 6 days post-infection. Analysis was performed using ImageJ software (version 1.52a) and increased intensity is directly proportional to the extent of leaf yellowing. Control indicates uninfected leaf tissue. **D.** CFU recovered from *A. thaliana* Col-0 leaves infiltrated with *Pto* DC3000 *hrcC/hrcN* complementation strains at 0, 2 and 3 DPI as indicated. **E.** Average pixel intensity analysis for *A. thaliana* Col-0 leaves infiltrated with *Pto* DC3000 complementation strains 6 days post-infection. Analysis was performed using ImageJ software (version 1.52a) and increased intensity is directly proportional to the extent of leaf yellowing. Control indicates uninfected leaf tissue. In each case, different *hrcC/hrcN* alleles are indicated on the X-axis. Error bars show standard error of the mean, and asterisks denote statistically significant differences from the WT/D0 (2 sample t-test) where '**' denotes $p = \leq 0.01$, '***' ('****' in **B**) denotes $p = \leq 0.001$. **A:** n = 4 plants; **B, D:** n = 3 plants; **C,E:** n = 8 leaves.
(TIF)

**S2 Fig. A.** CFU recovered from *A. thaliana* Col-0 leaves infiltrated with *Pto* DC3000 *wspR19*-expressing strains at 0, 2 and 3 DPI as indicated. **B.** Average pixel intensity analysis for *A. thaliana* Col-0 leaves infiltrated with *Pto* DC3000 *wspR19*-expressing strains 6 days post-infection. Analysis was performed using ImageJ software (version 1.52a) and increased intensity is directly proportional to the extent of leaf yellowing. **C.** CFU recovered from *A. thaliana* Col-0 leaves infiltrated with *Pto* DC3000 *bifA*-expressing strains at 0, 2 and 3 DPI as indicated. **D.** Average pixel intensity analysis for *A. thaliana* Col-0 leaves infiltrated with *Pto* DC3000 *bifA*-expressing strains 6 days post-infection. Analysis was performed using ImageJ software (version 1.52a) and increased intensity is directly proportional to the extent of leaf yellowing. In each case, different *hrcC/hrcN* alleles are indicated on the X-axis. Error bars show standard error of the mean, and asterisks denote statistically significant differences from the WT/D0 (2 sample t-test) where '*' = $p \leq 0.05$ and '***' = $p \leq 0.001$. **A,C:** n = 3 plants; **B,D:** n = 8 leaves.
(TIF)

**S3 Fig. A.** CFU recovered from *A. thaliana bbc* (immunocompromised) leaves infiltrated with *Pto* DC3000 *hrcC/hrcN* mutants at 0, 2 and 3 DPI as indicated. **B.** Average pixel intensity analysis for *A. thaliana bbc* leaves infiltrated with *Pto* DC3000 *hrcC/hrcN* mutants 6 days post-infection. Analysis was performed using ImageJ software (version 1.52a) and increased intensity is directly proportional to the extent of leaf yellowing. **C.** CFU recovered from *A. thaliana fec* leaves infiltrated with *Pto* DC3000 *hrcC/hrcN* mutants at 0, 2 and 3 DPI as indicated. **D.** Average pixel intensity analysis for *A. thaliana fec* (immunocompromised) leaves infiltrated with *Pto* DC3000 *hrcC/hrcN* mutants 6 days post-infection. Analysis was performed using ImageJ software (version 1.52a) and increased intensity is directly proportional to the extent of leaf yellowing. In each case, different *hrcC/hrcN* alleles are indicated on the X-axis. Error bars show standard error of the mean, and asterisks denote statistically significant differences from the WT (2 sample t-test) where '***' denotes $p = \leq 0.001$. **A,C:** n = 3 plants; **B,D:** n = 8 leaves. **E.** Visual disease phenotypes of *Pto* DC3000 infiltrated *A. thaliana fec* leaves taken at random from 3 independent plants, 6 days post-infection. *Pto* strains harbour the *hrcC/hrcN* alleles as indicated.
(TIF)

**S4 Fig. A.** CFU recovered from *A. thaliana* Col-0 leaves infiltrated with *Pto* DC3000 effector overexpression strains at 0, 2 and 3 DPI as indicated. **B.** Average pixel intensity analysis for *A. thaliana* Col-0 leaves infiltrated with *Pto* DC3000 effector overexpression strains 6 days post-infection. Analysis was performed using ImageJ software (version 1.52a) and increased intensity is directly proportional to the extent of leaf yellowing. Control indicates uninfected leaf tissue. **C.** CFU recovered from *A. thaliana* Col-0 leaves infiltrated with *Pto* DC3000 effector deletion mutants at 0, 2 and 3 DPI as indicated. **D.** Average pixel intensity analysis for *A. thaliana* Col-0 leaves infiltrated with *Pto* DC3000 effector deletion mutants 6 days post-infection. Analysis was performed using ImageJ software (version 1.52a) and increased intensity is directly proportional to the extent of leaf yellowing. Control indicates uninfected leaf tissue. In each case, different *hrcN*/effector mutants are indicated on the

X-axis. Error bars show standard error of the mean, and asterisks denote statistically significant differences from the WT/D0 (2 sample t-test) where '*' = $p \leq 0.05$, '**' = $p \leq 0.01$, and '***' = $p \leq 0.001$. **A,C:** n = 3 plants; **B,D:** n = 8 leaves.
(TIF)

**S1 Data. Raw data. for all graphs contained in this manuscript.**
(ZIP)

## Acknowledgments

The authors would like to thank Eleftheria Trampari and Isabella Frost for insightful conversations about the project, and Lucia Grenga and Sebastian Pfeilmeier respectively for the pBBR4-*wspR19* and pBBR4-*bifA* plasmids.

## Author contributions

**Conceptualization:** Danny Ward, Jacob G. Malone.

**Formal analysis:** Danny Ward, Catriona M. A. Thompson.

**Funding acquisition:** Jacob G. Malone.

**Investigation:** Danny Ward, Richard H. Little, Catriona M. A. Thompson.

**Methodology:** Danny Ward.

**Project administration:** Jacob G. Malone.

**Supervision:** Richard H. Little, Jacob G. Malone.

**Writing – original draft:** Danny Ward, Jacob G. Malone.

**Writing – review & editing:** Danny Ward, Richard H. Little, Catriona M. A. Thompson, Jacob G. Malone.

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
