## [Decision Letter · Decision Letter 0]

24 Aug 2025

Cyclic-di-GMP controls Type III effector export and symptom development in Pseudomonas syringae infections via the export ATPase HrcN

PLOS Pathogens

Dear Dr. Malone,

Please submit your revised manuscript within 60 days Oct 23 2025 11:59PM. If you will need more time than this to complete your revisions, please reply to this message or contact the journal office at plospathogens@plos.org. Please include the following items when submitting your revised manuscript:

We look forward to receiving your revised manuscript.

Kind regards,

Sébastien Bontemps-Gallo

Academic Editor

PLOS Pathogens

Savithramma Dinesh-Kumar

Section Editor

Editor-in-Chief

PLOS Pathogens

orcid.org/0000-0003-2946-9497

Editor-in-Chief

PLOS Pathogens

orcid.org/0000-0002-7699-2064

**Journal Requirements:**

https://journals.plos.org/plospathogens/s/submission-guidelines#loc-parts-of-a-submission

4) We notice that your supplementary Figures, and Tables are included in the manuscript file. Please remove them and upload them with the file type 'Supporting Information'. Please ensure that each Supporting Information file has a legend listed in the manuscript after the references list.

Potential Copyright Issues:

i) Please confirm (a) that you are the photographer of 1C, 3, 5, and S3E, or (b) provide written permission from the photographer to publish the photo(s) under our CC BY 4.0 license.

6) We note that your Data Availability Statement is currently as follows: "All data used in this submission are contained in the main manuscript and supplementary information files.". Please confirm at this time whether or not your submission contains all raw data required to replicate the results of your study. Authors must share the “minimal data set” for their submission. PLOS defines the minimal data set to consist of the data required to replicate all study findings reported in the article, as well as related metadata and methods (https://journals.plos.org/plosone/s/data-availability#loc-minimal-data-set-definition).

7) Please amend your detailed Financial Disclosure statement. This is published with the article. It must therefore be completed in full sentences and contain the exact wording you wish to be published.

2) If any authors received a salary from any of your funders, please state which authors and which funders..

**Reviewers' Comments:**

Reviewer's Responses to Questions

**Part I - Summary**

Reviewer #1: This study investigates how the bacterial second messenger cyclic-di-GMP (cdG) regulates the Type III secretion system (T3SS) in Pseudomonas syringae, with a particular focus on the ATPase HrcN. Mutations in two residues (G176A and G311A), predicted to interact with cdG, result in attenuated plant disease symptoms while leaving bacterial proliferation in Arabidopsis intact. The authors convincingly decouple symptom development from in planta growth and link the phenotype to altered secretion of a subset of effectors, particularly HopAA1-2, which appears essential for symptom development. The manuscript is well written and the experimental framework is robust, combining genetic, biochemical, and plant infection assays. The use of effector-CyaA translocation assays and immune-compromised plant lines adds further depth. The study provides interesting insight into how second messengers such as cdG can post-translationally modulate T3SS function, an area of growing interest in bacterial pathogenesis.

However, the mechanistic interpretation remains somewhat speculative due to some limitations: the biochemical characterization of only one mutant (G176A), despite G311A showing an equally pronounced phenotype; and the observation that G176A still binds cdG despite its phenotype, which complicates the model that cdG binding per se is required for regulatory control. Additional experiments are needed to resolve whether mutations disrupt cdG binding affinity, downstream allosteric signaling, or HrcN complex stability. The study is timely and significant, but clarification of these mechanistic issues is important.

Reviewer #2: This work is based on previous results from the team showing that cdGMP binds to a type of ATPases which includes T3SS ATPase from Pseudomonas syringae HrcN. Following the rational that binging to the ATPase needed for such a key virulence determinant to work is expected to have biological consequences which have not bee previously explored. Bases on Alphaphold predictions, several residues predicted to be involved in cdGMP binding were selected for mutagenesis and characterization. These mutations do not alter bacterial colonization of plant tissues but some affect the appearance of associated disease symptoms leading to the authors proposing that their impact may be related to translocation of some but not all effectors, and their testing this by following translocation of 23 effectors.

The initial premise of this report is an interesting one since T3SS ATPases are indeed important for a critical virulence determinant and very little has been done to characterize their biological roles in plant pathogens. In fact, one issue that should be addressed by the authors is that in its current form, although the vast majority of the relevant knowledge comes from mechanistic and structural insight obtained from a few animal pathogens, this is often not clear in the text. Although it is correct to take that as working hypothesis for other T3SS, it is important to make clear that it has only been shown for that T3SS. Similarly, effector translocation is presented as coordinated to follow a time hierarchy but the only reference included is for Salmonella (ref 38 line 65). Specific mentions to plants do link different effectors to differential impacts on early or late infection events, however, although early events must imply early translocation, phenotypes associated to late events do not necessarily mean late effector translocation since the impact for an early molecular interaction may be only noticeable later on. Additionally, when this issued is presented again in the discussion section (144-145), it is again presented as a generally established fact, this time including two references, ref 38 from Salmonella as before, and a plant-linked reference (6) where this issue is mentioned as a future challenge not as a proven fact. These generalizations often lead to assumptions on aspects that have not been properly proven, and in this particular case, does not help to properly frame the relevance of their work, which is enhanced by the scarcity of reports on the mechanistic characterization of these systems beyond a few animal pathogens. Thus, the authors should thoroughly review the introduction and discussion to carefully address citations to avoid this type of issues.

Additionally interesting aspects of this report include the infrequent separation of impacts on symptom induction and bacterial proliferation found for mutations in HrcN residues selected on the basis of their potential relevance for cdGMP binding. Although common to other types of plant pathogens (e.g. viruses) most mutations affecting disease development in plant pathogenic bacteria affect bacterial colonization. The link to differential effector translocation, and in particular to the translocation of specific effectors that affect induction of symptoms without affecting bacterial colonization is another interesting novelty of the report. Altogether, the research presented is of wide interest to the field.

Reviewer #3: In this study, Ward and colleagues investigate how a putative interaction between cyclic diguanylate (c-di-GMP) and the HrcN ATPase. HcrN is a protein that interacts with the type 3 injectisome (T3I) – a molecular syringe that delivers toxic effectors to target host cells. The T3I is important for the virulence of many Pseudomonas species. This includes the plant pathogen Pseudomonas syringae that is the focus of this study.

The context of the research is well positioned relative to research with other Pseudomonas species and the interaction of c-di-GMP with ATPases in other systems. How c-di-GMP affect type 3 secretions systems is an open question in the field. The current study addresses that knowledge gap.

Through a series of genetic, biochemical and functional assays in Arabidopsis thaliana, the authors seek to demonstrate that c-di-GMP binding to HcrN adjusts the range of effector proteins secreted by the T3I, thereby influencing the outcome of plant infections. The results, if accurate, are an advance to fundamental knowledge of signal transduction that will be of interest to many in the field. However, I have a few concerns with the interpretation of data, and I’m not convinced that all the conclusions are justified. I hope the authors will consider and address this feedback.

**Part II – Major Issues: Key Experiments Required for Acceptance**

Reviewer #1: The DRaCALA binding assay results (Fig. 2B–C) suggest that both WT and G176A bind cdG, but the competition assays imply altered affinity in G176A. However, this is not quantified. ITC or other biochemical experiments is needed (or at least densitometric quantification of DRaCALA) to compare cdG binding affinities of WT and G176A. This is important to support one of the proposed mechanism that mutations modulate effector export via altered cdG-HrcN interactions.

Only G176A was characterized biochemically. G311A, which shows a similar infection phenotype, was not analyzed due to solubility issues. However, excluding it weakens the generalizability of the conclusions. In addition, it would be interesting to test additional mutants (e.g., E208D, L338V, F174Y) for cdG binding and ATPase activity to validate whether phenotypically neutral mutations retain biochemical function. Alternatively, attempt solubilization of G311A using different buffers, expression systems, or tags.

The authors show that only a subset of effectors are affected in G176A/G311A, but it is unclear whether this reflects post-transcriptional effects, altered recognition, or secretion docking.

Reviewer #2: -Mutant variants of HrcN are tested by Western blot for impact on T3SS assembly using antibodies to detect HrcN and HrcQ. However, no control for loading (normalization of the samples) is mentioned in either the text, the methods or the fig. legend. Since differences in band intensity are visible in the HrcN and HrcQ (in HrcQ and contaminating FliI) membranes, and a quantitative difference could be potentially relevant, such internal control and relative quantification is important.

-In addition, the fact that the proteins are detected in the infected-leaf samples does not mean that the T3SS apparatus is correctly assembled (lines 116-119), since the authors are not even testing subcellular locations (i.e. membrane bound fractions) but whole extracts, so their conclusions (once ratified by proper quantification) can include the absence of changes in protein production, not assembly. Results and discussion sections should be reviewed in relation to this issue.

-Also related to western blot results, the figure legend includes mention to FliI being detected as a contamination and the identity of all three proteins (HrcN, HrcQ and FliI) being established by mass spectrometry. However, no additional mention to this is found neither on the result nor the methods section. This should be included at least in the corresponding Western blot analysis section of the methods.

-In figures where images are shown to illustrate impact on symptom induction, figure legends include mention to how many leaves and plants are shown, which are described as representative, but lack reference to the final N analyzed. The method section only describes Image J analysis, and there the minimum N used is mentioned, but it is inconsistent at least in some cases with the numbers include in the figure legend. For example, Fig.1C is said to show 6 representative leaves from 4 plants, while the image J methods describe a minimum of 8 leaves per experiment. Are these 6 leaves representative of 8 leaves? Then why not show all? If representative of more, it should indicate so in the specific legend. Also, should indicate if 4 plants from the same or independent experiments. These issues should be fixed by including the specific N or each of these experiments in the specific legends for all figures showing this type of data.

-The results showing that mutations to the residues affecting symptom induction still bind cdGMP are puzzling. Since these mutant variants display altered dynamics, and the link to cdGMP is well established by overexpression of WspR, the relation between the phenotypes and cdGMP regulation appears to be there. However, this issue should be covered in the discussion as it is an important mechanistic aspect of the work that remains to be established. In relation to this, the results obtained after overexpression of BifA is also another puzzling result that should be properly discussed.

Reviewer #3: Lines 92-101 and Fig. 1. Could the authors provide more information about the validation of AlphaFold3 structures shown in Fig. 1? Currently a previous study from the group is cited; however, details of the previous structural elucidation are lacking. AlphaFold3 is powerful but imperfect. Additional context would be helpful so that readers don’t need to refer to the previous study to understand the current one. Also, more information in-text about the biochemistry/structure of the c-di-GMP binding site would be helpful because it is a central player in the story. Likewise, additional rationale for the engineered HcrN mutations could be included in the text. Can the authors provide additional clarification?

Line 130-131. In contrast to what is stated in the text, it appears as thought the ATPase function of HcrN-G176A may be at least partially compromised. Consider that the Vmax for WT appears to be higher than that of the G176A mutant, and it is possible that 1.5 mM ATP – a concentration at which activity for WT is greater than the mutant – may be a physiologically relevant GTP concentration in live cells. Could the authors more carefully justify/consider their statements about activity? I worry that this result might confound interpretation.

Line 131-137 and Fig. 2. The crux of the paper seems to revolve around c-di-GMP binding to HcrN; however, the HcrN-G176A mutant – which is studied extensively in the manuscript – does not seem exhibit impaired c-di-GMP binding. This observation is difficult to understand in the narrative of the manuscript. Is it possible that the DRACALA assay is misleading here? The results in Fig. 2 C are not as clear as one might expect based on previous descriptions of the DRACALA assay (e.g., see Fig. 2 in the original PNAS paper from Vincent Lee’s lab describing the assay, https://www.pnas.org/doi/10.1073/pnas.1018949108). In fact, most of the results for competition assays appear to affect c-di-GMP binding relative to the solo fluorescent-c-di-GMP spot situated above them. G176A appears diminished in intensity relative to wild type. The proposed mechanism relies on a clear result here – could the authors please use a complementary technique to verify these data? The use of complimentary, validating techniques is a standard for protein-protein interactions to validate results. It is a big ask – but isothermal titration calorimetry or another biophysical protein-ligand binding technique might help to clarify the interpretation. As presented, the data here seem a bit murky.

**Part III – Minor Issues: Editorial and Data Presentation Modifications**

Reviewer #1: Figure 1D:

Missing uninfected plant control, despite being mentioned in the text.

Absence of HrcN in ΔhrcC mutant needs explanation, does HrcN stability depend on the T3SS complex?

Band intensity in G176A appears weaker, possible instability or reduced expression? Adding a loading control (e.g., EF-Tu or RNApol) for normalization.

Figure 2:

Include a catalytically dead mutant as a negative control for ATPase activity.

DRaCALA data should be quantified. Fluorescence images alone are insufficient to assess subtle changes.

In Fig. 2C, unlabeled cdG only partially competes for G176A, this warrants quantification and further interpretation.

Figure 3:

The water control (described in line 151) is not shown in the figure.

Figures S2 and S3 relate directly to Fig. 3 and should be merged for clarity.

The conclusion that cdG levels affect symptom development should address why both high and low cdG restore virulence.

Figure 4:

In Fig. 4B, labels are inconsistent, WT and G176A are mentioned instead of G311A.

Authors claim 5 effectors increased and 4 decreased in G311A, but only one each show statistical significance.

ΔhrcN control is mentioned but not shown in 4A or 4B.

Effector order in panels A and B should be identical to facilitate comparison.

Legend is under-detailed: specify time post-infection, n values, statistical tests, and normalization methods.

Discrepancy between HopAF1 in Fig. 4A (non-significant) and Fig. 4C (significant) needs clarification.

Why was HopAF1 not tested in 4D, despite its altered secretion profile in other panels?

Figure 5:

In 5A, HopH1 overexpression is phenotypic but not deleted in 5B why?

Reviewer #2: -Also related to individual data used for the figures, all graphs show individual data as well as mean and variation. Such an increasingly used practice provides clarity and granularity to the data, and it is important as its helps to interpret results while contributing to avoid reproducibility issues.

-Complementation assays from Tn7 expression are very nice! Assays using plant defense mutants, as well as those including effector mutants are also appreciated

Reviewer #3: Overall, the manuscript is short and well written. However, there are a large number of abbreviations (e.g., Pto, T3I, T3E, etc.), and I wonder if some of the abbreviations might be removed. I leave this to the discretion of the authors.

What is WspR19?

Fig. 1, panel D. I respect the truth in advertising; however, is there a possibility to increase specificity of the antiserum by using an adsorption protocol (thereby eliminating some of the extraneous bands)?

The fonts in Fig. S4 need to be adjusted – some are so large that they appear to overlap, others are so small that they are nearly illegible. Could the authors use a consistent, easily legible font size?

PLOS authors have the option to publish the peer review history of their article (what does this mean? ). If published, this will include your full peer review and any attached files.

**Do you want your identity to be public for this peer review?** For information about this choice, including consent withdrawal, please see our Privacy Policy .

Reviewer #1: No

Reviewer #2: No

Reviewer #3: **Yes: ** Joe J. Harrison

**Figure resubmission:**

**Reproducibility:**



---

## [Decision Letter · Decision Letter 1]

14 Dec 2025

PPATHOGENS-D-25-01712R1

Cyclic-di-GMP controls Type III effector export and symptom development in Pseudomonas syringae infections via the export ATPase HrcN

PLOS Pathogens

Dear Dr. Malone,

Thank you for submitting your manuscript to PLOS Pathogens. Your manuscript is almost ready to be accepted. Could you please resubmit a version in which the acronym PDE is defined at its first appearance.

After careful consideration, we feel that it has merit but does not fully meet PLOS Pathogens's publication criteria as it currently stands. Therefore, we invite you to submit a revised version of the manuscript that addresses the points raised during the review process.

We look forward to receiving your revised manuscript.

Kind regards,

Sébastien Bontemps-Gallo

Academic Editor

PLOS Pathogens

Savithramma Dinesh-Kumar

Section Editor

PLOS Pathogens

Sumita Bhaduri-McIntosh

Editor-in-Chief

PLOS Pathogens

orcid.org/0000-0003-2946-9497

Michael Malim

Editor-in-Chief

PLOS Pathogens

orcid.org/0000-0002-7699-2064

**Journal Requirements:**

**Reviewers' Comments:**

Reviewer's Responses to Questions

**Part I - Summary**

Reviewer #1: This manuscript is substantially improved, with clearer framing of the biological findings, improved organization, and a more balanced presentation of the study’s mechanistic claims.

Reviewer #2: As described in my previous review, this is an interesting report of significance and novelty to the field, and after revision no major concerns remained.

Reviewer #3: The authors have made a considerable effort to address reviewer comments. This included executing protein production and biophysical assessments of ligand binding. I recognize that this represented significant effort, even if the results are easily interpreted due to technical limitations. The authors have done a nice job of clarifying text and narrative with respect to novelty of the study in plant pathogenesis. Limitations are acknowledged in a substantially revised discussion section. I am supportive of publication.

**Part II – Major Issues: Key Experiments Required for Acceptance**

Reviewer #1: (No Response)

Reviewer #2: No major issues remains in the revised version

Reviewer #3: None.

**Part III – Minor Issues: Editorial and Data Presentation Modifications**

Reviewer #1: (No Response)

Reviewer #2: In their response to reviewers comments, the authors state that not only Pto, T3I, T3E, and cdG acronysms have been maintained in the revised manuscript, but in the discussion a couple on sentences have kept PDE twice without having been spelled out previously: lines 284-289.

Reviewer #3: None.

PLOS authors have the option to publish the peer review history of their article (what does this mean? ). If published, this will include your full peer review and any attached files.

**Do you want your identity to be public for this peer review?** For information about this choice, including consent withdrawal, please see our Privacy Policy .

Reviewer #1: No

Reviewer #2: **Yes: ** Carmen R Beuzon

Reviewer #3: No

**Figure resubmission:**
---

## [Editor Report · Decision Letter 2]

17 Dec 2025

Dear Dr Malone,

We are pleased to inform you that your manuscript 'Cyclic-di-GMP controls Type III effector export and symptom development in Pseudomonas syringae infections via the export ATPase HrcN' has been provisionally accepted for publication in PLOS Pathogens.

Best regards,

Sébastien Bontemps-Gallo

Academic Editor

PLOS Pathogens

Bart Thomma

Section Editor

PLOS Pathogens

Sumita Bhaduri-McIntosh

Editor-in-Chief

PLOS Pathogens

orcid.org/0000-0003-2946-9497

Michael Malim

Editor-in-Chief

PLOS Pathogens

orcid.org/0000-0002-7699-2064
---

## [Editor Report · Acceptance letter]

Dear Dr Malone,

We are delighted to inform you that your manuscript, " 

Cyclic-di-GMP controls Type III effector export and symptom development in Pseudomonas syringae infections via the export ATPase HrcN," has been formally accepted for publication in PLOS Pathogens.

Best regards,

Sumita Bhaduri-McIntosh

Editor-in-Chief

PLOS Pathogens

orcid.org/0000-0003-2946-9497

Michael Malim

Editor-in-Chief

PLOS Pathogens

orcid.org/0000-0002-7699-2064